# Antimicrobial Hexaaquacopper(II) Complexes with Novel Polyiodide Chains

**DOI:** 10.3390/polym13071005

**Published:** 2021-03-24

**Authors:** Zehra Edis, Radhika Raheja, Samir Haj Bloukh, Richie R. Bhandare, Hamid Abu Sara, Guido J. Reiss

**Affiliations:** 1Department of Pharmaceutical Sciences, College of Pharmacy and Health Science, Ajman University, PO Box 346, Ajman, United Arab Emirates; r.bhandareh@ajman.ac.ae; 2Center of Medical and Bio-Allied Health Sciences Research, Ajman University, Ajman, United Arab Emirates; s.bloukh@ajman.ac.ae (S.H.B.); h.abusara@ajman.ac.ae (H.A.S.); 3SVKM’S Dr. Bhanuben Nanavati College of Pharmacy, Mumbai 400056, India; radhika.raheja@bncp.ac.in; 4Department of Clinical Sciences, College of Pharmacy and Health Science, Ajman University, PO Box 346, Ajman, United Arab Emirates; 5Institut fur Anorganische Chemie und Strukturchemie, Heinrich-Heine University Düsseldorf, 40225 Düsseldorf, Germany; guido.reiss@uni-duesseldorf.de

**Keywords:** antimicrobial agents, polymeric pentaiodides, copper, crown-ether, halogen bonding, crystal structure

## Abstract

The non-toxic inorganic antimicrobial agents iodine (I_2_) and copper (Cu) are interesting alternatives for biocidal applications. Iodine is broad-spectrum antimicrobial agent but its use is overshadowed by compound instability, uncontrolled iodine release and short-term effectiveness. These disadvantages can be reduced by forming complex-stabilized, polymeric polyiodides. In a facile, in-vitro synthesis we prepared the copper-pentaiodide complex [Cu(H_2_O)_6_(12-crown-4)_5_]I_6_ · 2I_2_, investigated its structure and antimicrobial properties. The chemical structure of the compound has been verified. We used agar well and disc-diffusion method assays against nine microbial reference strains in comparison to common antibiotics. The stable complex revealed excellent inhibition zones against *C. albicans WDCM 00054*, and strong antibacterial activities against several pathogens. [Cu(H_2_O)_6_(12-crown-4)_5_]I_6_ · 2I_2_ is a strong antimicrobial agent with an interesting crystal structure consisting of complexes located on an inversion center and surrounded by six 12-crown-4 molecules forming a cationic substructure. The six 12-crown-4 molecules form hydrogen bonds with the central Cu(H_2_O)_6_. The anionic substructure is a halogen bonded polymer which is formed by formal I_5_^−^ repetition units. The topology of this chain-type polyiodide is unique. The I_5_^−^ repetition units can be understood as a triodide anion connected to two iodine molecules.

## 1. Introduction

Mankind faces dangerously rising levels of antimicrobial resistance by the so-called ESKAPE pathogens (*Enterococcus faecium, Staphylococcus aureus, Klebsiella pneumoniae, Acinetobacter baumannii, Pseudomonas aeruginosa, Enterobacter* spp. and *Escherichia coli*) [1,2]. Conventional drugs and antimicrobials lose their efficiency against such emerging multi-drug resistant microorganisms [2,3,4,5]. Pathogens can be acquired in hospital settings through nosocomial infections and lead to delayed recovery, treatment failures, increasing health care costs, morbidity, and mortality [5,6]. In the current COVID-19 pandemic, nosocomial infections originating from emergency rooms and health care settings have negatively impacted the treatment of immunocompromised, severely ill patients with comorbidities [7,8].

Antimicrobial polymeric coatings can reduce the burden of infections through contaminated fomites in all indoor and outdoor settings [8,9,10,11,12,13]. Inclusion of known antimicrobial agents like iodine and copper into polymeric coating materials can mitigate antimicrobial resistance [13,14,15,16].

Iodine forms interesting polyiodide structures depending on the given stabilizing and complexing agents [17,18,19,20,21,22,23,24,25,26,27,28,29,30]. Polyiodides are formed from basic [I_2k+n_]^n−^ units (k = integer, n = 1–4), that can be controlled through synthesis [17,18]. Extended polyiodide networks result through attachment of triiodide I_3_^−^anions (I_3_^−^) to iodine (I_2_) molecules by halogen and hydrogen bonding [11,17,18,19,20,21,22,23,24,25,26]. 1,4,7,10-tetraoxacyclododecane (12-crown-4), a crown-ether, forms stable tri- and pentaiodides [27,28,29,30,31]. The crown-ether molecules stabilize the anionic structure around sandwiched central metal cations [27,28,29,30,31,32].

The use of iodine and copper in one compound can ameliorate antimicrobial activities due to synergism. The complex [Cu(12-crown-4)_5_(H_2_O)_6_]I_6_ · 2I_2_ contains both inorganic biocides and is expected to show inhibitory action on pathogens. We tested our compound against a total of 10 microbial strains in comparison to five common antibiotics. The inhibitory effect of our complex polymeric compound on *C. albicans* WDCM 00054 Vitroids showcased its antifungal activity. [Cu(12-crown-4)_5_(H_2_O)_6_]I_6_ · 2I_2_ exhibits excellent activity against reference strains of microorganisms compared to selected antibiotics. The methods used for the characterization of our novel polymeric complex compound [Cu(12-crown-4)_5_(H_2_O)_6_]I_6_ · 2I_2_ confirmed its composition and structure. The results are in accordance to previous studies [27,28,29,30,31]. The complex consists of sandwiched copper-hexahydrates within polymeric pentaiodide-chains. A recent database check revealed that the interesting topology of this chain-type polyiodide is new.

## 2. Materials and Methods

### 2.1. Chemicals

Iodine (≥99.0%), copper iodide (CuI) and Mueller Hinton Broth (MHB) were obtained from Sigma Aldrich (Gillingham, UK). 1,4,7,10-Tetraoxacyclododecan (12-crown-4) was received from Sigma-Aldrich Chemical Co. (St. Louis, MO, USA). Disposable sterilized Petri dishes with Mueller Hinton II agar and McFarland standard sets were bought from Liofilchem Diagnostici (Roseto degli Abruzzi (TE), Italy). The bacterial strains *E. coli* WDCM 00013 Vitroids, *P. aeruginosa* WDCM 00026 Vitroids, *K. pneumoniae* WDCM00097 Vitroids, *C. albicans* WDCM 00054 Vitroids*,* and *Bacillus subtilis* WDCM0003 Vitroids were received from Sigma-Aldrich Chemical Co. *S. pneumoniae* ATCC 49619, *S. aureus* ATCC 25923, *E. faecalis* ATCC 29212, *S. pyogenes ATCC 19615, P. mirabilis ATCC 29906* were purchased from Liofilchem. Gentamicin (9125, 30 µg/disc), cefotaxime (9017, 30 µg/disc), chloramphenicol (9128, 10 µg/disc), streptomycin (SD031, 10 µg/disc), and nystatin (9078, 100 IU/disc) were obtained from Liofilchem. Methanol (analytical grade) was received from EMSURE (Merck KGaA, Darmstadt, Germany). Acetonitrile (analytical grade) was purchased from MTEDIA (TEDIA Company, Fairfield, OH, USA). Sterile filter paper discs with a diameter of 6 mm were bought from Himedia (Mumbai, India). Ultrapure water was utilized and all reagents were of analytical grade and used as received.

### 2.2. Preparation of [Cu(H_2_O)_6_(12-crown-4)_5_]I_6_ · 2I_2_

CuI (0.12 g, 0.63 mmol) is dissolved within 10 min in acetonitrile (10 mL) in a 250 mL beaker covered with Parafilm under stirring at 35 °C. In another beaker covered with Parafilm, I_2_ (0.32 g, 1.26 mmol) is dissolved in methanol (30 mL) under stirring at room temperature. After completely dissolving the reagents, the two solutions are mixed under continuous stirring at 35 °C for 5 min. Under stirring 12-crown-4 (0.2 mL, 1.26 mmol) is then added dropwise. Stirring is continued until the solution is clear and everything is dissolved. The crystals appear after 3–5 days at room temperature through slow evaporation. The dark crystals are filtrated and render a percentage yield of 73%.

### 2.3. Characterization of the Compound

The compound was investigated by several methods. Microstructural analysis was done by FE-SEM, the purity of the compound was verified by EDS. UV-Vis, FTIR and Raman spectroscopy affirmed the composition and functional groups. X-ray crystallography revealed the structure of the compound. These methods confirmed the composition and structure of our pentaiodide.

#### 2.3.1. Electron Microscope (SEM) and Energy-Dispersive X-ray Spectroscopic (EDS) Analysis

The sample imaging for the morphological characteristics and the analysis for the composition of the compound was conducted by FE-SEM (NOVA Nano FESEM 450, FEI, Hillsboro, Oregon, USA) at 15 kV. The surface morphology, elemental EDS information and composition details were obtained by using two different modes. The low vacuum mode measurement was performed with an uncoated sample while in the high vacuum mode the sample was coated with Au/Pd using a sputter coater.

#### 2.3.2. Nuclear Magnetic Resonance (NMR) Spectroscopy

The ^1^H- and the ^13^C- nuclear magnetic resonance (NMR) measurements were done on a Bruker (Billerica, MA, USA) instrument with DMSO-d_6_ at a frequency of 400 MHz.

#### 2.3.3. Characterization by Raman Spectroscopy (Raman)

The Raman analysis was done by a RENISHAW (Gloucestershire, UK) system equipped with an optical microscope at room temperature, a 532 nm solid-state laser beam in the spectral range of 100–3000 cm^−1^, output power 15% and a spectral resolution of −1 cm^−1^ with the integration time of −300 s. The excitation of the solid-state laser beam was focused on the sample through the 50× objective of a confocal microscope with a spot diameter of —2 μm). The sample solution was placed into a standard cuvette (1 cm × 1 cm) on the pathway of the laser beam. The CCD-based monochromator collected and analyzed the scattered light.

#### 2.3.4. Fourier-Transform Infrared (FTIR) Spectroscopy

The compound was analyzed from 400–4500 cm^−1^ with a FTIR spectrometer (Shimadzu, Kyoto, Japan). FTIR analysis was utilized to verify the structural components of the molecule.

#### 2.3.5. Crystal Structure Analysis

Data collection was performed on an Xcalibur2 diffractometer (Rigaku, Tokyo, Japan) using the standard procedures implemented in the CrysAlis^PRO^ software [33]. The radiation type was Mo-K_α_ with a wavelength of 0.71073. The diffractometer has a KM4 collision model CCD plate. Empirical absorption correction was done by using spherical harmonics, implemented in SCALE3 ABSPACK scaling algorithm. The structure was solved using intrinsic phasing [34] and refined using the SHELXL software [35,36]. Hydrogen atoms were added using the standard procedures for riding models implemented in the SHELX program. The triiodide anion shows a small disorder at I5, which was refined with a fixed ratio of 4:1, which fits best with the reduction of the difference electrons density in this area. All figures showing the title structure are drawn using the DIAMOND software [37]. CCDC 2065523 contains the supplementary crystallographic data for this paper. These data can be obtained free of charge from The Cambridge Crystallographic Data Centre via www.ccdc.cam.ac.uk/structures. (Accessed on 15 March 2021).

### 2.4. Investigation/Determination of Antibacterial and Antifungal Properties of [Cu(H_2_O)_6_(12-crown-4)_5_]I_6_ · 2I_2_

[Cu(H_2_O)_6_(12-crown-4)_5_]I_6_ · 2I_2_ was tested on the Gram-positive bacteria *S. pneumonia* ATCC 49619*, S. aureus* ATCC 25923*, S. pyogenes* ATCC 19615, *E. faecalis* ATCC 29212, *B. subtilis* WDCM0003 and the Gram-negative bacteria *E. coli* WDCM 00013 Vitroids*, P. mirabilis* ATCC 29906*, P. aeruginosa* WDCM 00026 Vitroids and *K. pneumonia* WDCM00097 Vitroids. The antifungal activity of our complex polymeric compound was tested against *C. albicans* WDCM 00054 Vitroids.

#### 2.4.1. Bacterial Strains and Culturing

The reference strains of Gram-positive *S. pneumoniae* ATCC 49619, *S. aureus* ATCC 25923, *E. faecalis* ATCC 29212, S*. pyogenes* ATCC 19615, *Bacillus subtilis* WDCM0003 Vitroids, Gram-negative *P. mirabilis* ATCC 29906, *E. coli* WDCM 00013 Vitroids, *P. aeruginosa* WDCM 00026 Vitroids, *K. pneumoniae* WDCM00097 Vitroids, and fungus *C. albicans* WDCM 00054 Vitroids were used for the zone of inhibition plate studies. These strains were stored at −20 °C, inoculated in Mueller Hinton Broth (MHB) by adding the respective fresh bacteria and fungi into 10 mL the MHB and stored at 4 °C until further use.

#### 2.4.2. Procedure for Zone of Inhibition (ZOI) Plate Studies

The susceptibilities of the selected 10 reference strains against [Cu(H_2_O)_6_(12-crown-4)_5_]I_6_ · 2I_2_ were investigated by the zone of inhibition plate method. The complex is not completely soluble in polar solvents, therefore we utilized the method described by Kirby-Bauer [38]. The pathogens were suspended in 10 mL Mueller Hinton Broth (MHB). Bacteria were incubated for two to four hours at 37 °C. Fungi were incubated at 30 °C on Sabouraud dextrose broth. Then the microbial culture were adjusted to 0.5 McFarland standard. In each case, 100 μL was seeded uniformly with sterile cotton swabs on disposable, sterilized Petri dishes with Mueller Hinton II agar (MHA). The agar plates were dried for 10 min and utilized for agar well diffusion and disc diffusion essays.

#### 2.4.3. Agar Well Diffusion Method

A sterile well borer was used to remove 6 mm diameter circular pieces of MHB agar from already inoculated agar plates. These wells were filled with 20 mg [Cu(H_2_O)_6_(12-crown-4)_5_]I_6_ · 2I_2_, and the negative controls methanol (75 µL) and acetonitrile (75 µL). These prepared plates with the bacterial reference strains and *C. albicans* were then incubated for 24 h at 37 °C and 30 °C, respectively. A ruler was utilized to measure the diameter of the zone of inhibition (ZOI) to the nearest millimeter. The susceptibility of the selected reference strains are revealed by the diameters of clear ZOI around the well. The absence of any clear ZOI around the wells indicates microbial resistance. The agar well diffusion assays were done three times and the results are the average of these three independent experiments.

#### 2.4.4. Disc Diffusion Method

Sterile filter paper discs (Himedia) with a diameter of 6 mm were soaked with 2 mL of known concentrations of the stock solutions of [Cu(H_2_O)_6_(12-crown-4)_5_]I_6_ · 2I_2_ in methanol/acetonitrile (50 µg/mL (1), 2 mL of 25 µg/mL (2) and 2 mL of 12.5 µg/mL (3)) These impregnated discs were dried at room temperature for 24 h. The negative control discs were prepared by impregnating filter paper discs with 2 mL of the solvents methanol and acetonitrile and drying them at room temperature for 24 h.

The antimicrobial activity of the reference strains against our compound, the negative control solvents, and the positive control antibiotics gentamicin, streptomycin, amikacin, cefotaxime, chloramphenicol and nystatin were tested by disk diffusion method according to recommendations of the Clinical and Laboratory Standards Institute (CLSI) [39]. The prepared discs were placed on the previously inoculated agar plates. Bacterial strains were incubated at 37 °C for 24 h, while the agar plates with *C. albicans* were incubated at 30 °C for 24 h. A ruler was used to measure the diameter of the ZOI to the nearest millimeter. The susceptibility of the selected reference strains are revealed by the diameters of clear ZOI around the well. The absence of any clear ZOI around the wells indicates microbial resistance. The agar well diffusion assays were done three times and the results are the average of these three independent experiments.

### 2.5. Statistical Analysis

Data were expressed as mean deviation. The statistical significance between groups was calculated with one-way ANOVA. A value of *p* < 0.05 was considered as statistically significant. The statistical analysis was done by using the SPSS software (version 17.0, SPSS Inc., Chicago, IL, USA).

## 3. Results and Discussion

### 3.1. Elemental Composition and Morphological Examination

#### Electron Microscope (SEM) and Energy-Dispersive X-ray Spectroscopic (EDS) Analysis

The morphology and composition of [Cu(H_2_O)_6_(12-crown-4)_5_]I_6_ · 2I_2_ were investigated through SEM and EDS analysis (Figure 1, Appendix A).

The SEM analysis at 15 kV in high-vacuum mode shows heterogenous morphology of the crystalline sample (Figure 1). The EDS information confirms the purity of the sample and the composition of our compound (Appendix A). The sample holder is made of copper, this increased the weight percent of Cu (Appendix A).

### 3.2. Spectroscopical Characterization

#### 3.2.1. Nuclear Magnetic Resonance (NMR) Spectroscopy

The ^13^C- and the ^1^H- nuclear magnetic resonance (NMR) measurement results are in accordance with previous results for similar 12-crown-4-polyiodide complexes, with singlets at 70.40 ppm and 3.521 ppm, respectively (Appendix A) [28,30,32].

The signals originate from the methylene groups within the 12-crown-4 molecule. Pure 12-crown-4 exhibits signals at 3.435 and 70.387 ppm in the ^1^H- and ^13^C-NMR spectra, respectively (Appendix A). The ^1^H chemical shifts for [Cu(H_2_O)_6_(12-crown-4)_5_]I_6_ · 2I_2_ are similarly downfield like our previously reported compounds [M(12-crown-4)_2_]I_x_ (M = Li, Na, K, Rb, Cs and x = 3, 5, 7) in comparison to the pure 12-crown-4 (Appendix A) [28,30,32]. The signals are at low field due to the complexation in the compounds. The strain on the 12-crown-4 molecules leads to deshielding of the nuclei and causes a paramagnetic shift (Appendix A). The deshielding in [Cu(H_2_O)_6_(12-crown-4)_5_]I_6_ · 2I_2_ is smaller than in the sandwiched [M(12-crown-4)_2_]I_x_ complexes [28,30,32]. In the structure of [Cu(H_2_O)_6_(12-crown-4)_5_]I_6_ · 2I_2_ the five 12-crown-4-ethers experience less strain within the structure and appear in high field compared to our previously investigated complexes [28,30,32]. The same reasons govern the ^13^C-NMR analytical results and are coherent to previous results for 12-crown-4 compounds (Appendix A) [28,30,32].

#### 3.2.2. Raman Spectroscopy (Raman)

The Raman analysis confirms partly the composition of [Cu(H_2_O)_6_(12-crown-4)_5_]I_6_ · 2I_2_ and is verified by similar results in previous studies (Figure 2) [28,32].

The range until 200 cm^−1^ is used to identify the topology of polyiodides. The bands below this important range cannot be clearly assigned due to high luminescence except for two peaks at 172 and 142 cm^−1^. The absorption at 172 cm^−1^ originates from the I_2_-units within [Cu(H_2_O)_6_(12-crown-4)_5_]I_6_ · 2I_2_ in agreement with other studies (Table 1) [18,20,21,32,40,41,42].

The peak at 142 cm^−1^ corresponds to the asymmetric stretching mode in I_5_^−^ -units within the polymeric chain and is identified as such in other studies as well (Figure 2, Table 1) [20,43,44]. Asymmetric triiodide stretch modes appear usually between 130–140 cm^−1^ and cannot be clearly detected here. The strong absorption of the iodine molecules, overlap with 12-crown-4 absorptions. Our technical limitations resulted in high luminescence and prevented further verifications [20,22].

As a result, the Raman shifts confirm the topology of the polymeric-chain pentaiodides consisting of two iodine molecules. The triiodide units could not be assigned due to ambiguity caused by the high luminescence in this part of the spectrum. The spectrum shows a peak at 2855 cm^−1^, which verifies the presence of 12-crown-4 molecules within the complex compound (Appendix A) [32].

#### 3.2.3. Fourier-Transform Infrared (FTIR) Spectroscopy

FTIR analysis was utilized to verify the structural components of [Cu(H_2_O)_6_(12-crown-4)_5_]I_6_ · 2I_2_ (Appendix A). The results are in agreement with our previously published 12-crown-4-polyiodide complexes [M(12-crown-4)_2_]I_x_ [28,30,32]. Table 2 illustrates the bands in the FTIR spectrum of pure 12-crown-4 and [Cu(H_2_O)_6_(12-crown-4)_5_]I_6_ · 2I_2_.

[M(12-crown-4)_2_]I_x_ and [Cu(H_2_O)_6_(12-crown-4)_5_]I_6_ · 2I_2_ show similar absorptionbands in the FTIR spectrum related to the crown ether (Table 2, Appendix A) [28,30,32]. The signals confirm the structure of [Cu(H_2_O)_6_(12-crown-4)_5_]I_6_ · 2I_2_ (Table 2). Symmetric, asymmetric vibrational stretching and deformation bands for the crown ether molecule are available in the FTIR spectrum of [Cu(H_2_O)_6_(12-crown-4)_5_]I_6_ · 2I_2_ (Appendix A). These are the absorption bands of C-H, CH_2_, C-C, C-O, and CH-CH [28,30,32]. The spectrum of [Cu(H_2_O)_6_(12-crown-4)_5_]I_6_ · 2I_2_ contains absorption signals indicating the presence of hydroxyl groups (Table 2, Appendix A) [11,28,30,32]. The signal at 3746 and the broad band around 3418 cm^−1^ are due to symmetric (ν_1s_) and asymmetric (ν_2a_) stretching vibrations of hydroxyl groups, respectively (Table 2, Appendix A) [11,45]. These two bands show a blue shift in comparison to corresponding pure water vibrational stretching modes at 3418 and 3200 cm^−1^ [45]. The shift to higher energy absorption in [Cu(H_2_O)_6_(12-crown-4)_5_]I_6_ · 2I_2_ means decreased hydrogen bond formation in comparison to pure water [45]. The band at 1642 cm^−1^ originates from the H-O-H bending mode ν_3_ of free water molecules and confirms the hydrating effect of H_2_O molecules within [Cu(H_2_O)_6_(12-crown-4)_5_]I_6_ · 2I_2_ (Table 2, Appendix A) [45]. The absorption of this vibrational bending mode appears at lower energy compared to the corresponding bending mode absorption of pure water at 1650 cm^−1^ [45]. This red shift confirms again decreased hydrogen bond formation within [Cu(H_2_O)_6_(12-crown-4)_5_]I_6_ · 2I_2_ in comparison to the hydrogen bond network in water [45]. The absorption bands at 628 and 549 cm^−1^ are related to librational modes, which appear at 600−800 cm^−1^ in the FTIR spectrum of water [45].

### 3.3. Cyrstal Structure Determination

#### Crystal Structure Analysis

The crystal structure analysis of [Cu(H_2_O)_6_(12-crown-4)_5_]I_6_ · 2I_2_ revealed a novel topology. Details of the crystal data and the refinement are listed in Table 3.

The asymmetric unit of the title crystal structure [Cu(H_2_O)_6_(12-crown-4)_5_]I_6_ · 2I_2_ contains one half of a hexaaquacopper(II) complex, 2.5 12-crown-4 molecules, two halves of an iodine molecule and one triiodide anion (Figure 3).

In detail, the hexaaquacopper(II) complex is located on an inversion center (Wyckoff site 1f). The Cu-O bond lengths are in accordance with a typical hexaaqua copper complex showing a Jan-Teller stretching (Cu-O1W: 1.961(4) Å; Cu-O2W: 1.973(3) Å; Cu-O3W: 2.298(3) Å). Each hexaaquacopper(II) complex is surrounded by six 12-crown-4 molecules, which are connected via 12 O-H^...^O hydrogen bonds (Figure 5). These units are furthermore connected to two neighboring hexaaquacopper(II) complexes by two more O-H^...^O hydrogen bonds.

As aforementioned there are 2.5 12-crown-4 molecules in the asymmetric unit. Two crystallographically independent molecules are located in general positions whereas one half of a 12-crown-4 molecule is locaed around an inversion center (Wyckoff site 1b). The 12-crown-4 molecule showing this inversion symmetry forms the connection to the adjacent heaxaaquacopper(II) complex (Figure 3). Consequenty all three crystallographically independent 12-crown-4 molecules show different conformations to fill the needs of bonding and packing forces. An interesting feature of this structure is the presence of all three prominent conformers of the 12-crown-4 molecule. In one of the molecules, all four oxygen atoms are on the same side of the ring, in the second one there are three oxygen atom at one side, and in the centrosymmetrical one its two to two.

The anionic part of the title structure is a chain-type polymer consisting of triiodide anions and iodine molecules. Each triiodide anion forms two halogen bonds to neighboring iodine molecules by one iodine atom (I3). The bond lengths within the formal triiodide anion (I3–I4–I5) are very asymmetric (I3–I4 = 3.0563(5) Å, I4–I5A = 2.8014(10) Å) [25,26]. Thus an alternative description would be an iodide anion (I3) surrounded by three iodine molecules. But the secondary halogen bonds to I1 and I2 are with I^...^I distances of 3.3780(6) Å for I1–I3 and 3.3781(7) Å I2–I3 significantly longer and obviously much weaker (Figure 4) [46,47,48,49].

The two weakly bonded iodine molecules are arranged around inversion centers (Wyckoff sites: 1c and 1g). Summing up all the geometric parameters, the description as an asymmetric I_3_^−^ anion bonded to two iodine molecules is the best choice. The presence of very asymmetric triiodide anions is well known and may be caused by hydrogen bonds [25] and by halogen bonds [26]. The mean plane of all atoms in this polyiodide chain is parallel to the *bc* plane, but the moieties are folded in a typical manner. The branched chain runs along the c direction (Figure 4).

In general, unbranched polyiodide chains are well knowns for decades [18]. In the case of short-chain units the halogen bonded interaction is realised by a head-to-tail connection. For the most simple polyiodide anion [I_3_^−^]_∞_ the cases of branched chains (not head-to-tail connected) are rare [50]. Polymeric chains with repetition units of I_5_^−^ are well known. But in most cases the the I_5_^−^ units are head-to tail connected similar to the [I_3_^−^]_∞_ [49]. A recently performed search in the Cambridge Structural database yielded no entry for the very unusual chain-topology presented in this contribution. For this database search we used a I_3_^−...^ I_2_^...^ I_3_^−...^ I_2_ moiety with the I_3_^−^ anions connected to two iodine molecules by one of its iodine atoms (Figure 4). A polyiodide salt that shows some structural similarity concerning its connectivity to the polyiodide presented here is a I_18_^2−^ polyiodide anion seen in the structure of [M(phen)]I_18_ (M=Fe, Ni) [51]

The two subsystems of the title structure discussed before excellently fit with each other. A more general view on the the title structure shows, that the hexaaquacopper(II) complex is located in the center of the unit cell directly together with four 12-crown-4 molecules. The 12-crown-4 molecule and the anionic polyiodide substructure are located in the *bc* plane. (Figure 5). The title structure obviously features medium strong hydrogen bonds that connects the hexaaquacopper(II) complexes and the 12-crown-4 molecules. These connections as well as packing forces—like van der Waals forces—determine the different conformations of the 12-crown-4 molecules in the cationic substructure and the orientation of the water ligands. The polyiodide substructure perfectly fits into this structure and there are only weak interactions between the cationic and the anionic substructure that contribute to the stability of the whole structure.

### 3.4. Determination of Antimicrobial Activities of [Cu(12-crown-4)_5_(H_2_O)_6_]I_6_ · 2I_2_

[Cu(12-crown-4)_5_(H_2_O)_6_]I_6_ · 2I_2_ was tested on 10 different pathogens in comparison to positive control antibiotics gentamicin, streptomycin, amikacin, erythromycin, cefotaxime, and chloramphenicol (Table 4).

The susceptibility of the selected microorganisms against our compound were confirmed by dilution series, agar well and disc diffusion methods. All results of the negative controls methanol and acetonitrile showed no zone of inhibition (ZOI) and were not mentioned in Table 4. The results are comparable to our previous results with sandwiched 12-crown-4-complexes [27,29,31].

The hexaaquacopper(II) complex inhibits the fungal strain *C. albicans WDCM 00054* strongly and proves to be more effective than the antibiotic nystatin (Table 3, Figure 6a). Our polymeric iodide-chain-complex shows stronger antibacterial activity against the Gram-positive strains *S. aureus ATCC 25923* and *B. subtilis WDCM 0003* in comparison to the antibiotics gentamicin and streptomycin, respectively (Table 3, Figure 6b,c). The complex inhibits the Gram-negative pathogens *E. coli WDCM 00013* and *K. pneumoniae WDCM 00097* stronger than the respective antibiotics amikacin and cefotaxime (Table 3, Figure 6d,e).

The agar well studies reveal strong microbicidal action of [Cu(12-crown-4)_5_(H_2_O)_6_]I_6_ · 2I_2_ against most of the reference strains (Table 3, Figure 7). The strongest antimicrobial activity is documented against *C. albicans WDCM 00054* (Table 3, Figure 7a). The Gram-positive strains *S. aureus ATCC 25923*, *S. pneumoniae ATCC 49619* and *E. faecalis ATCC 29212* show strong to intermediate inhibition zones (Table 3, Figure 7b–d). The Gram-negative pathogens *P. aeruginosa WDCM 00026* and *E. coli WDCM 00013* are inhibited intermediately by our hexaaquacopper(II) complex (Table 3, Figure 7d,e).

The polymeric compound [Cu(12-crown-4)_5_(H_2_O)_6_]I_6_ · 2I_2_ exhibits remarkable microbicidal action against the used reference strains compared to common antibiotics. In AW studies and DD studies *C. albicans WDCM 00054* showed the highest susceptibility towards our polyiodide, followed by *S. aureus ATTC 25932* (Table 3). In general, the AW studies revealed highest inhibitory action against the fungus *C. albicans WDCM 00054*, followed by Gram-positive and lastly Gram-negative pathogens (Figure 7). This order is different in disc diffusion studies due to higher susceptibility of Gram-negative *E. coli WDCM 00013*, and *K. pneumoniae WDCM 00097* towards our complexed polymer (Table 2, Figure 6). [[Cu(12-crown-4)_5_(H_2_O)_6_]I_6_ · 2I_2_ shows in DD studies the strongest antimicrobial activity against *C. albicans WDCM 00054* (ZOI = 53)*,* followed by the Gram-positive *S. aureus ATTC 25932* (ZOI = 35), *B. subtilis WDCM 00003* (ZOI = 33), then the two Gram-negative *E. coli WDCM 00013* (ZOI = 25), and *K. pneumoniae WDCM 00097* (ZOI = 24) (Figure 6). After the latter, the order is restored with Gram-positive *S. pyogenes ATCC 19615* (ZOI = 21), *S. pneumoniae ATCC 49619* (ZOI = 18), *E. faecalis ATCC 29212* (ZOI = 18), followed by Gram-negative *P. mirabilis ATCC 29906* (ZOI = 15), and *P. aeruginosa WDCM 00026* (ZOI = 12) (Table 3). The two Gram-negative pathogens *E. coli WDCM 00013* and *K. pneumoniae WDCM 00097* are higher susceptible in disc diffusion studies compared to agar well methods.

The inhibitory action of [Cu(12-crown-4)_5_(H_2_O)_6_]I_6_ · 2I_2_ shows a general increasing trend starting from Gram-negative to Gram-positive bacteria and culminates in strong antifungal activity against the fungus *C. albicans WDCM 00054.* Gram-negative pathogens have higher negatively charged cell surfaces and are less susceptible towards [Cu(12-crown-4)_5_(H_2_O)_6_]I_6_ · 2I_2_. They have a complicated outer cell membrane structure with inner and outer leaflet [52]. The outer leaflet contains lipopolysaccharides, which result in higher negative charge on the cell surface and can interact with positively charged groups or atoms [52]. This outer membrane forms a strong barrier against antimicrobial agents and is followed by a thin layer of peptidoglycan without inclusions of teichoic acid (TA) and lipoteichoic acid (LTA) [52]. Gram-positive species have a thick layer of peptidoglycan with TA and LTA inclusions. The peptidoglycan structure is a mesh-like structure crosslinked by peptide interbridges with partially negative oxygen atoms which can interact with partially positive groups or atoms. Gram-negative microorganisms have less crosslinking and additional porin channels in the inner leaflets [52]. These porin channels allow passive diffusion of small, charged, hydrophilic species [52]. Gram-positive strains and fungi lack porin channels as entry points within their cell membranes [52]. The high antibacterial activity of [Cu(12-crown-4)_5_(H_2_O)_6_]I_6_ · 2I_2_ against Gram-positive pathogens suggests in agreement with previous reports an interaction of the partial positively charged carbon atoms in 12-crown-4 molecules of our complex polymer with the partial negatively charged oxygen atoms in the peptide interbridges within the peptidoglycan layer [52]. This dipole-dipole interaction leads to a deformation of the [Cu(12-crown-4)_5_(H_2_O)_6_]I_6_ · 2I_2_ structure and results in the release of iodine molecules from the polymeric iodide chains [52]. Iodine molecules are strong antimicrobial agents and immediately exert their action on the cell membranes by different mechanisms [52]. Iodine molecules penetrate through the bacterial cell membranes by passive diffusion, enter the bacterial cytoplasm causing protein oxidation and inactivation of efflux pumps [52,53,54]. The latter is important for the inhibition of the Gram-negative ESKAPE pathogens *E. coli*, *P. aeruginosa* and *K. pneumoniae* [54]. These multi-drug resistant strains have efflux pumps which eliminate toxic compounds from periplasm and cytoplasm [54]. Our hexaaquacopper(II) complex is effective in disabling their efflux pumps due to iodine. Another mechanism is the iodination of the double bonded carbon atoms within the cell membrane fatty acids [53]. This action destroys the cell membrane and leads to microbial death through inhibition of metabolic pathways and membrane rupture [52].

The morphology and form of aggregation of the microorganisms is a strong indicator for the inhibitory action of our polymeric complex compound. The susceptibility of cocci is highest in *S. aureus*, followed by *S. pyogenes*, then *S. pneumonia* and finally *E. faecalis*. These microorganisms appear in form of clusters, chains, pairs and single, in pairs or chains, respectively. Staphylococci are more inhibited by [Cu(12-crown-4)_5_(H_2_O)_6_]I_6_ · 2I_2_ than streptococci. Non-motile species are more susceptible to our compound than motile strains with flaggellae.

## 4. Conclusions

In this study we investigated the structure, composition, morphology and antimicrobial activity of a novel hexaaquacopper(II) complex with polymeric pentaiodide chains. All analytical results are in agreement with previous studies of related compounds. The title compound [Cu(12-crown-4)_5_(H_2_O)_6_]I_6_ · 2I_2_ reveals an interesting structure, which consists of three different conformers of 12-crown-4-molecules. The unique topology of the polymeric polyiodide chains facilitate the antimicrobial activity against pathogens by controlled iodine release from its polymeric I_2_-I_3_^−^-I_2_ chain structure.

[Cu(12-crown-4)_5_(H_2_O)_6_]I_6_ · 2I_2_ has excellent antifungal properties against *C. albicans WDCM 00054* and inhibits strongly the studied bacterial strains. These results suggest the use of hexaaquacopper(II) complexes with polymeric-pentaiodide chains as antimicrobial coating agents against resistant pathogens causing nosocomial infections.

## Figures and Tables

**Figure 1 polymers-13-01005-f001:**
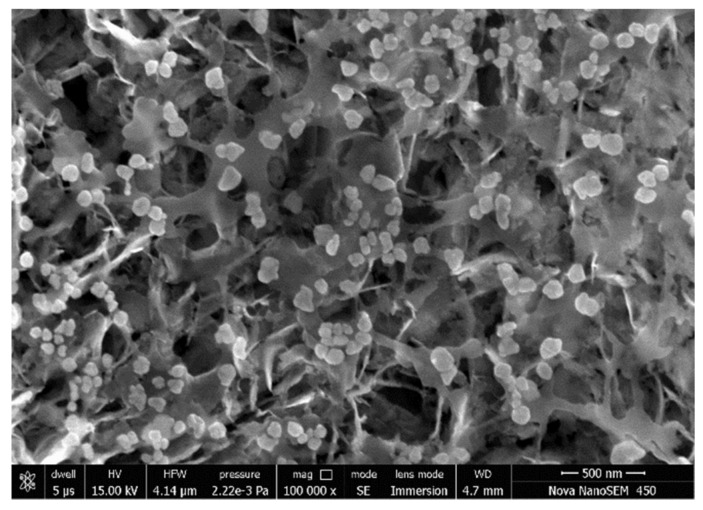
Scanning electron microscopy (SEM) of [Cu(12-crown-4)_5_(H_2_O)_6_]I_6_ · 2I_2_.

**Figure 2 polymers-13-01005-f002:**
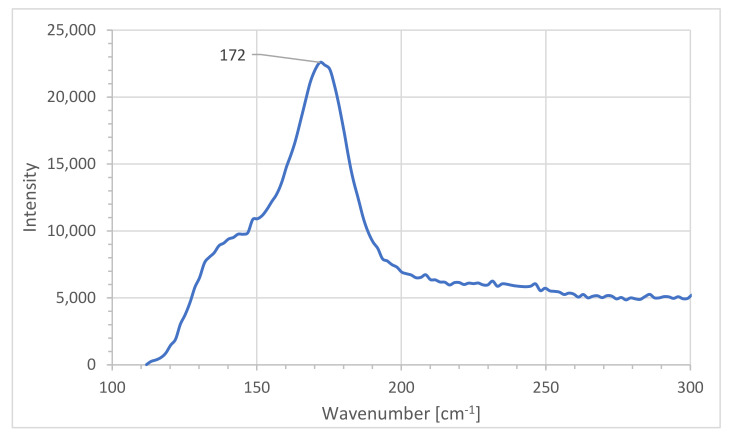
Raman spectroscopic analysis of [Cu(12-crown-4)_5_(H_2_O)_6_]I_6_ · 2I_2_.

**Figure 3 polymers-13-01005-f003:**
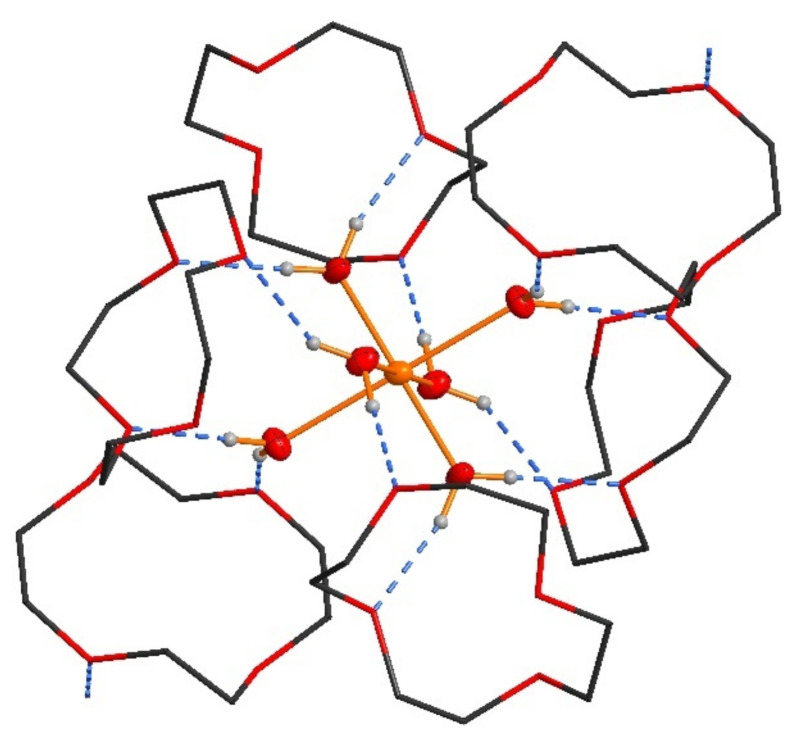
[Cu(12-crown-4)_5_(H_2_O)_6_]I_6_ · 2I_2_. Hexaaquacopper(II) complex surrounded by six 12-crown-4 molecules. The non-hydrogen atoms of the hexaaquacopper(II) complex are shown as ellipsoids with a probability of 50%. The 12-crown-4 molecules are shown in wireframe style.

**Figure 4 polymers-13-01005-f004:**
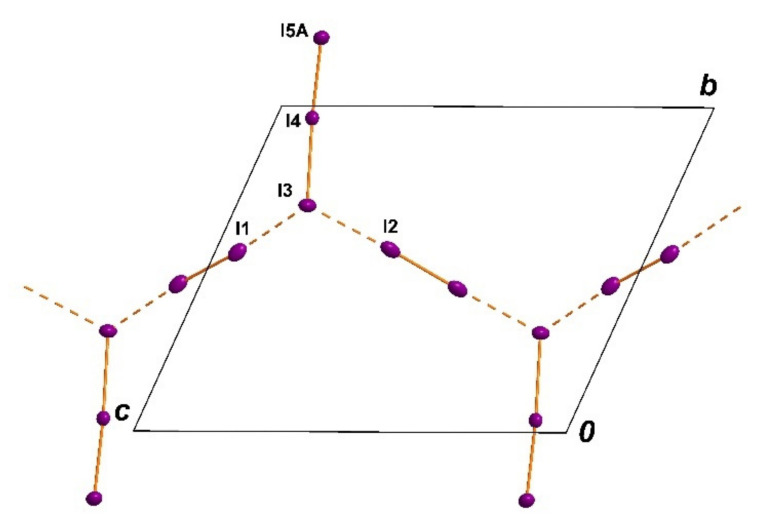
[Cu(12-crown-4)_5_(H_2_O)_6_]I_6_ · 2I_2_. All shown atoms are drawn with a 50% probability of the displacement ellipsoids. The branched chain is located parallel to the *bc* plane running along the *c* direction.

**Figure 5 polymers-13-01005-f005:**
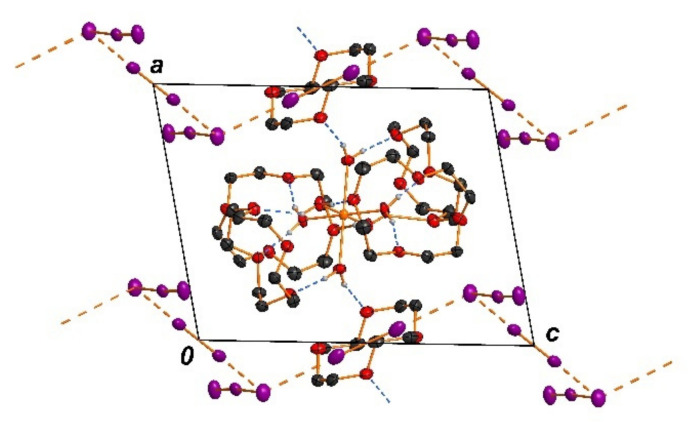
[Cu(12-crown-4)_5_(H_2_O)_6_]I_6_ · 2I_2_. Showing the interlock between hexaaquacopper(II) complex encapsulated by 12-crown-4 molecules (center of the unit cell) and the polymeric polyiodide substructure (*bc* plane).

**Figure 6 polymers-13-01005-f006:**
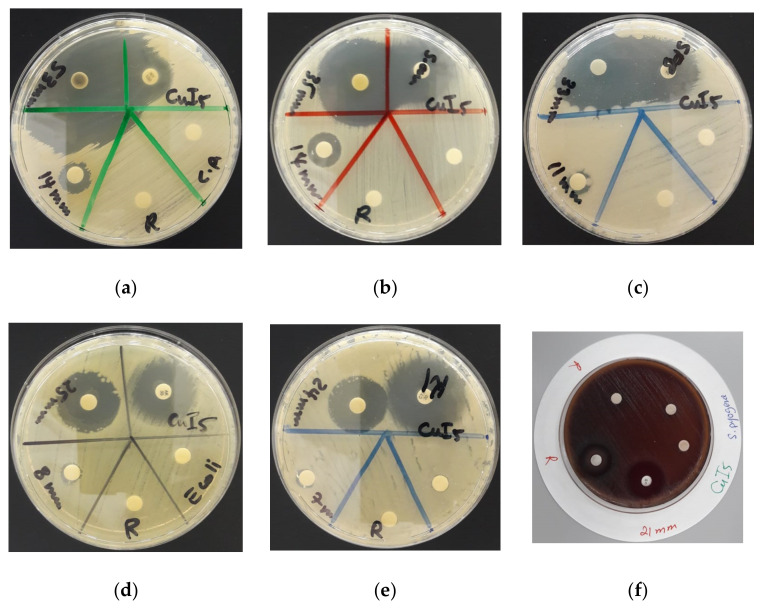
Disc diffusion methods of [Cu(12-crown-4)_5_(H_2_O)_6_]I_6_ · 2I_2_ with positive controls (antibiotic). From left to right: [Cu(12-crown-4)_5_(H_2_O)_6_]I_6_ · 2I_2_ against (**a**) *C. albicans WDCM 00054*; (**b**) *S. aureus ATCC 25932;* (**c**) *B. subtilis* WDCM 00003; (**d**) *E. coli* WDCM 00013; (**e**) *K. pneumoniae WDCM 00097*; (**f**) S. pyogenes *ATCC 19615*.

**Figure 7 polymers-13-01005-f007:**
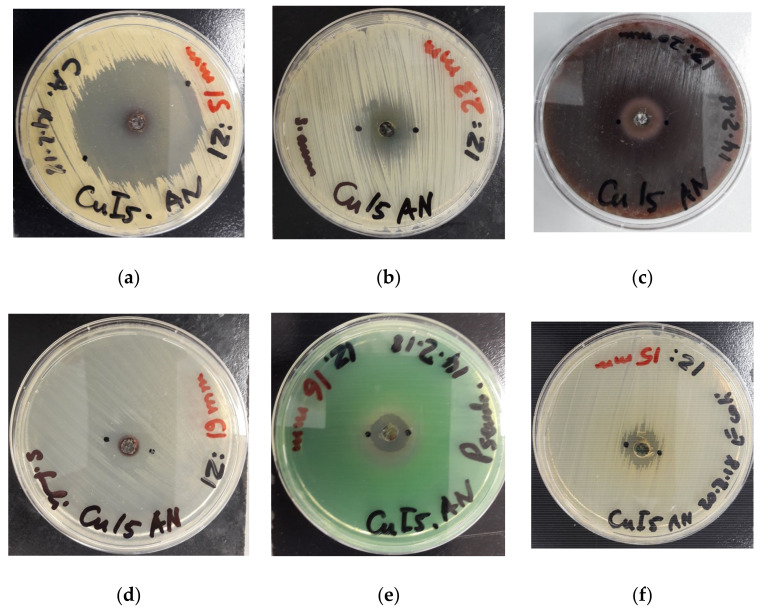
Agar well methods of [Cu(12-crown-4)_5_(H_2_O)_6_]I_6_ · 2I_2_. From left to right: [Cu(12-crown-4)_5_(H_2_O)_6_]I_6_ · 2I_2_ against (**a**) *C. albicans WDCM 00054*; (**b**) *S. aureus ATCC 25932;* (**c**) S. pyogenes ATCC 19615; (**d**) *E. faecalis ATCC 29212*; (**e**) *P. aeruginosa WDCM 00026*; (**f**) *E. coli WDCM 00013*.

**Table 1 polymers-13-01005-t001:** Raman shifts of iodine moieties in the samples [Cu(H_2_O)_6_(12-crown-4)_5_]I_6_ · 2I_2_ (1) (cm^−1^).

Group	1	[43]	[44]	[22]	[20]
I_2_	172 ν_s_		172 ν_s_		172 ν_s_
I_3_^−^		217 ν_as_ 331 ν_as_		227 ν_as_ 340 ν_as_	
I_5_^−^	142 ν_as_	147 ν_as_	137 ν_as_		134 ν_as_
12-crown-4	2855				

ν = vibrational stretching, s = symmetric, a = asymmetric.

**Table 2 polymers-13-01005-t002:** FTIR analysis [cm^–1^] of 12-crown-4 (A) [28,30,32] and [Cu(12-crown-4)_5_(H_2_O)_6_]I_6_ · 2I_2_ (B).

	ν_1_ (O–H) * _s_ ν_2_ (O–H) * _a_	ν (C–H)_a_	ν (CH_2_)_a,s_	ν (C-H)_s_ ν (O-H) *	δ (C-H)_a_ ν_3_ (H-O-H) *	(C-C)	δ (C-H) δ (C-C)	ν (C-O)	ν (CH-CH) ν (O-H) *
A		2940	2909	2860	1459	1375	1290	1250 1140 1100 1025 919	848
B	3746 * _s_ 3418 * _a_	2951	2905	2863 2777 * 2733 *	1443 1642 *	1360	1287	1243 1133 1092 1022 911	844 549 * 628 *

* Signals related to vibrational modes of hydroxyl groups due to hydration in the compound B. ν = vibrational stretching, δ = deformation, s = symmetric, a = asymmetric.

**Table 3 polymers-13-01005-t003:** Crystal data, data collection and refinement of [Cu(H_2_O)_6_(12-crown-4)_5_]I_6_ · 2I_2_.

Item	Parameter
Formula	C_40_H_92_CuI_10_O_26_
*Mr*	1160.84
Linear absorption factor	µ = 4.64 mm^−1^
Crystal system, space group	Triclinic , *P* *-1*
*a*	10.7289 (4) Å
*b*	12.3645 (5) Å
*c*	15.1570 (7) Å
α	113.470 (4)°
β	99.187 (4)°
γ	92.543 (3)°
Temp.	100 K
Volume, *Z*	*V* = 1807.70 (14) Å^3^, *Z* = 1
Diffractometer	Xcalibur
Radiation	Mo *K*α, λ = 0.71073 Å
Measured reflections	17,571
Independent reflections	7846
Reflections with *I* > 2σ(*I*)	5930
*R*int; Completeness	0.042; 99.3%
Refined parameters	371
*R*[*F*^2^ > 2σ(*F*^2^)]	0.042
*wR*(*F*^2^); GooF	0.081, 1.05
Δρ_max_; Δρ_min_	0.86 e Å^−3^; −0.87 e Å^−3^

**Table 4 polymers-13-01005-t004:** Antimicrobial testing of antibiotics (A), [Cu(12-crown-4)_5_(H_2_O)_6_]I_6_ · 2I_2_ by agar well (AW), and disc dilution studies (1,2,3). ZOI (mm) against microbial strains by diffusion assay.

Strain	Antibiotic	A	AW ^+^	1 ^+^	2 ^+^	3 ^+^
*S. pneumoniae ATCC 49619*	G	18	20	19	0	0
*S. aureus ATCC 25923*	G	28	23	35	14	0
*S. pyogenes ATCC 19615*	C	25	20	21	0	0
*E. faecalis ATCC 29212*	CTX	25	19	18	0	0
*B. subtilis WDCM 00003*	S	20	21	33	11	0
*P. mirabilis ATCC 29906*	G	25	0	15	0	0
*P. aeruginosa WDCM 00026*	CTX	21	16	12	0	0
*E. coli WDCM 00013*	A	20	15	25	8	0
*K. pneumoniae WDCM 00097*	CTX	17	NA	24	7	0
*C. albicans WDCM 00054*	NY	16	51	53	14	0

^+^ Agar well (AW) diffusion studies (20 mg crystals of [Cu(12-crown-4)_5_(H_2_O)_6_]I_6_ · 2I_2_ in 6 mm diameter well) and disc diffusion studies (6 mm disc impregnated with 2 mL of 50 µg/mL (1), 2 mL of 25 µg/mL (2) and 2 mL of 12.5 µg/mL (3) of [Cu(12-crown-4)_5_(H_2_O)_6_]I_6_ · 2I_2_). A Amikacin (30 µg/disc). G Gentamicin (30 µg/disc). CTX (Cefotaxime) (30 µg/disc). NY (Nystatin) (100 IU). C Chloramphenicol (10 µg/disc). Streptomycin (10 µg/disc). Grey shaded area represents Gram-negative bacteria. 0 = Resistant. No statistically significant differences (*p* > 0.05) between row-based values through Pearson correlation.

## Data Availability

CCDC 2065523 contains the supplementary crystallographic data for this paper. These data can be obtained free of charge from The Cambridge Crystallographic Data Centre via www.ccdc.cam.ac.uk/structures (accessed on 15 March 2021).

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
