# Peer review of "Antimicrobial Hexaaquacopper(II) Complexes with Novel Polyiodide Chains"

_polymers, 2021, doi:10.3390/polym13071005_

Round 1
Reviewer 1 Report
The manuscript could be accepted for publication in Polymers after major revisions. At the moment the manuscript is not interesting to read and should be briefed. Duplication between the text, figures, and tables should be avoided. For example, the following comments should be answered.
1) Line 20-22: the sentence should be simplified and briefed. For example, it could read: “The chemical structure of the compound has been verified”.
2) Page 5: The energy dispersive spectrum (Figure 1b) can be moved to the supplementary material.
3) Page 6: The NMR spectra (Figure 2) should be moved to the supplementary material.
4) Page 8: The FTIR data has been provided in details in table 2 and represented in Figure 4. Remove Figure 4 and move to the supplementary material.
5) Page 9: Move Table 3 to the supplementary material.
6) Page 13: Move Figure 8 to the supplementary material since the data is presented in Table 4.
7) Page 14: Move Figure 9 to the supplementary material.
8) Line 506: Correct the error “[]”. Either remove the brackets or add the reference number.
9) The conclusion should be briefed and simplified. It should concentrate in the most important findings of the study.

Author Response
Dear Reviewer,
thank you for your valuable and sincere comment.
Please see the attachment.
Best regards
Zehra
Comments and Suggestions for Authors
The manuscript could be accepted for publication in Polymers after major revisions. At the moment the manuscript is not interesting to read and should be briefed. Duplication between the text, figures, and tables should be avoided. For example, the following comments should be answered.
1) Line 20-22: the sentence should be simplified and briefed. For example, it could read: “The chemical structure of the compound has been verified”. True, thank you very much. Agreed and we shortened the sentence.
2) Page 5: The energy dispersive spectrum (Figure 1b) can be moved to the supplementary material. Agreed and moved.
3) Page 6: The NMR spectra (Figure 2) should be moved to the supplementary material. Agreed and moved. At the same time, more explanation added:” Opposite to the previously investigated complexes [M(12-crown-4)2]Ix, the title compound shows a similar 13C –shift like pure 12-crown-4 with 70.40 ppm compared to 70.387 ppm, respectively [28,30,32]. This result verifies, that the 12-crown-4 molecules in of [Cu(H2O)6(12-crown-4)5]I6 x 2I2 are less deformed by the complexation. The 13C-NMR spectra of the sandwiched complexes [M(12-crown-4)2]Ix reveal signals in the range below 70 ppm [28,30,32].” In the lines 253-258.
4) Page 8: The FTIR data has been provided in details in table 2 and represented in Figure 4. Remove Figure 4 and move to the supplementary material. Agreed and moved.
5) Page 9: Move Table 3 to the supplementary material. Dear Reviewer, we like to keep this table in the text, because it offers for the reader, who is interested in structures an immediate glimpse on the data, if you allow. If you insist, we can do it. Actually, the co-author and expert in crystal structure analysis, Dr. Reiss, will like to keep this in the text. I hope, you do not mind.
6) Page 13: Move Figure 8 to the supplementary material since the data is presented in Table 4. Dear Reviewer we agree, that it could be moved to the supplemental material, but it offers those readers, who are interested the materialized results of disc diffusion. We find it informative in terms of comparing the different techniques of AW and DD. At the same time, the reader can see the ZOI, the effect of the title compound on the different microorganisms in comparison to the antibiotics. Our co-authors, the microbiologists Dr. Samir Haj Bloukh and Mr Hamed Abu Sara are asking to keep the picures, which shows their work. If you do not mind, we keep them.
7) Page 14: Move Figure 9 to the supplementary material. Same problem here, with the above explanations. Please allow us to keep these parts in the manuscript.
8) Line 506: Correct the error “[]”. Either remove the brackets or add the reference number. Corrected, Reference number 53 was missing.
9) The conclusion should be briefed and simplified. It should concentrate in the most important findings of the study. I removed all additional explanations and concentrated on the major findings. Thank you so much.
Dear Reviewer, thank you so much for your valuable input. We tried our best to improve the manuscript according to your comments. We moved most of the items required into the supplemental material. Few item are left, which are related to our experts results. The manuscript is a joint work between microbiologists, inorganic chemists, pharmacists and especially x-ray crystallograpic experts. In this sense, it was not easy to ask moving the related results to the supplemental material, because everyone likes to see his/her hard work presented in the publication.
Thank you so much
Best regards
Dr. Zehra Edis
13/3/2021

Reviewer 2 Report
Dear Editors and Authors,
The Manuscript is well-written, the graphical aspect is illustrative, the scope of experimental methods meets the tasks of investigation.
Here are the comments, questions and considerations concerning the reviewed article:
- Why “Polymeric” in the article title is written with the capital letter?
- Term “microbicide” in general means not the same thing as the authors mean by it, see for example, the explanation from the WHO site.
https://www.who.int/hiv/topics/microbicides/microbicides/en/
- x-ray generally is written with the capital letter;
- “The anionic substructure is a halogen bonded polymer which is formed by formal I5 - repetition units. The topology of this chain-type polyiodide is unique.’’ In my opinion the authors should add a more detailed discussion what they mean by “unique topology” in the corresponding crystal structure characterization section.
- [I2k+n] n- - the authors should specify the possible values of k and n;
- Extended polyiodide networks result through attachment of triiodide I3—anions (I3 - ) to iodine (I2) molecules by halogen and hydrogen bonding – in my opinion, the role of hydrogen bond is doubtful in this case as the authors address here only the anionic part and thus the role of halogen bonding, dispersion or electrostatic interactions is dominant.
- What does the term “Vitroids” mean?
- In my opinion the fragment “The use of iodine and copper in one compound can ameliorate antimicrobial activi- 57 ties due to synergism. The complex [Cu(12-crown-4)5(H2O)6]I6xI2 contains both inorganic 58 biocides and is expected to show inhibitory action on pathogens. We tested our com- 59 pound against Gram-positive S. pneumonia ATCC 49619, S. aureus ATCC 25923, S. py- 60 ogenes ATCC 19615, E. faecalis ATCC 29212 and B. subtilis WDCM0003. E. coli WDCM 61 00013 Vitroids, P. mirabilis ATCC 29906, P. aeruginosa WDCM 00026 Vitroids and K. 62 pneumonia WDCM00097 Vitroids were used as Gram-negative bacterial strains. The inhibitory effect of our complex polymeric compound on C. albicans WDCM 00054 Vitroids 64 showcased its antifungal activity. These studies were conducted by agar well diffusion 65 (AW) and disc diffusion (DD) studies. The latter consisted of dilution series in comparison to the common antibiotics gentamicin, streptomycin, cefotaxime, chloramphenicol 67 and nystatin. [Cu(12-crown-4)5(H2O)6]I6xI2 exhibits excellent activity against reference 68 strains of microorganisms compared to selected antibiotics” in several different forms is repeated too much in the manuscript. The detailed experimental procedure should be specified in the experimental section and should not be duplicated anywhere else.
- The complex consists of sandwiched copper-hexahydrates within polymeric pentaiodide-chains. A recent database check revealed, that the interesting topology of this chain-type polyiodide is new. – A detailed address of the CSD (citation is also absent!) version and search details should be specified so that the readers and reviewers can test what the authors mean by “unique topology”.
- cupper iodide – incorrect spelling
- X-ray diffraction (XRD) and x-ray crystallography – X-ray with the capital letter and also the fragment may be considered wordy: one or the other part can be omitted.
- Mo-Ka with a wavelength of 0.71073 – the unit of length is absent
- Hydrogen atoms were addded – incorrect spelling
- [Cu(H2O)6(12-crown-4)5]I6 x 2I2 – no upper and lower text indices
- The 1H- and the 13C- nuclear magnetic resonance (NMR) measurements are in accordance with previous results for similar 12-crown-4-polyiodide complexes with singlets at 70.40 ppm and 3.521 ppm, respectively (Figure 2) [28,30,32]. – seems that the ppm values are given inconsistently: at the beginning of the phrase the sequence is 1H – 13C and the shifts are given so that the 13C values are in the first place.
- In my opinion, Fig. 2 should be better replaced to the supplementary section.
- Many times in the text there is a spelling "12-Krone-4, [M(12-Krone-4)2]Ix" which in my opinion is incorrect.
- a peak at 2855 – the unit is absent
- Raman spectrum should be rescaled or a break should be inserted in the medium-wavenumber region, as the main bands discussed by the authors are located below 200 cm-1 and thus interpretation is ambiguous. As the authors kindly provided their raw data, I was able to analyze it carefully. Unfortunately, the laser length used for spectrum excitation induced high luminescence in the most important range. I understand that in many cases the choice of the experimental setup is limited, but still, according to the quality of the spectrum, the authors can make reliably only the assignment of the band of bound iodine. The attribution of the bands at 232 and 351 cm-1 to overtones when we do not see initial bands (which should be intense according to the literature data) maybe too risky – we can also deal with the librations of crown part and anion…cation with respect to each other. Thus, I would recommend replacing Table 1 to the supplementary section and rewriting this part.
- What does the line “12-crown-4 2.891” in table 1 mean?
- What is the meaning of the red color in table 2?
- δ (C–C)s - in general the term symmetric-asymmetric is attributed to stretching vibrations, not deformational ones;
- The crystal structure analysis of [Cu(H2O)6(12-crown-4)5]I6 x 2I2 revealed a novel topology – details of CSD analysis should be added.
- Figure strukt 1 – unknown notation;
- Cu-O3WÅ: 2.298(3) – the unit is not in its place
- Figure 6. [Cu(12-crown-4)5(H2O)6]I6xI2. All shown atoms are drawn with a 50% probabilty of the 374 displacement ellipsids. The branched chain is located parallel to the bc plane running along the c 375 direction – typo in the "ellipsoid" spelling
- I3- anion – no upper-lower indices
- The presences - typo
- [49 and references cited there] – unknown citation
- The less negatively charged fungal and Gram-positive bacterial species – the phrase is misleading, I guess the authors mean membrane potential.
- membrane fatty acids [] – empty citation
Overall, all the comments above do not significantly lower the impression from the manuscript. As a suggestion for the future, it can be very interesting to study the processes of iodine release in details: kinetics, thermal stability, the energy of binding, the stability of powder samples on storage and in solutions, and many other thermodynamic and kinetic parameters that can be interesting from fundamental, physico-chemical and applied bactericidal point of view.
Author Response
Dear Reviewer,
tahnk you for your sincere and supportive comments.
Best regards
Zehra
Please see attachment or read below:
Here are the comments, questions and considerations concerning the reviewed article:
- Why “Polymeric” in the article title is written with the capital letter? True, thank you. Changed.
- Term “microbicide” in general means not the same thing as the authors mean by it, see for example, the explanation from the WHO site.
https://www.who.int/hiv/topics/microbicides/microbicides/en/
Dear Reviewer, thank you for your interesting comment. Iodine is a known antibacterial agent, which has bacteriocidal and fungicidal effects. This is the reason we used the word microbicide, which is a general term in shortened form. The page of the WHO is related to a specific case and explains the treatment needed. To avoid confusing readers, we replaced “microbicide” with the two words “antimicrobial agent” in the text.
As such, there were 6 uses of the word.
Abstract line 16, replaced with “broad-spectrum antimicrobial agent”.
Abstract line 24, replaced with “antimicrobial agent”
Line 46, replaced with “antimicrobial agents”
Last paragraph of the Results and Discussion, replaced with “antimicrobial agent”
- x-ray generally is written with the capital letter; True, sorry. (in paragraph of 2.3. Characterization of the compound)
- “The anionic substructure is a halogen bonded polymer which is formed by formal I5 - repetition units. The topology of this chain-type polyiodide is unique.’’ In my opinion the authors should add a more detailed discussion what they mean by “unique topology” in the corresponding crystal structure characterization section. Dear reviewer, thank you for your important comment. We added into the main text exactly these explanations regarding the Database search and the topology (lines 378-399).
- [I2k+n] n- - the authors should specify the possible values of k and n; True: I2k+n]n- units (k, n > 0; n = 1-4) line 50
- Extended polyiodide networks result through attachment of triiodide I3—anions (I3 - ) to iodine (I2) molecules by halogen and hydrogen bonding – in my opinion, the role of hydrogen bond is doubtful in this case as the authors address here only the anionic part and thus the role of halogen bonding, dispersion or electrostatic interactions is dominant. Thank you so much again. Please refer also to the added explanations on lines (378-399).
- What does the term “Vitroids” mean? It is the name of the reference strain, which is used by the company.
- In my opinion the fragment “The use of iodine and copper in one compound can ameliorate antimicrobial activi- 57 ties due to synergism. The complex [Cu(12-crown-4)5(H2O)6]I6xI2 contains both inorganic 58 biocides and is expected to show inhibitory action on pathogens. We tested our com- 59 pound against Gram-positive S. pneumonia ATCC 49619, S. aureus ATCC 25923, S. py- 60 ogenes ATCC 19615, E. faecalis ATCC 29212 and B. subtilis WDCM0003. E. coli WDCM 61 00013 Vitroids, P. mirabilis ATCC 29906, P. aeruginosa WDCM 00026 Vitroids and K. 62 pneumonia WDCM00097 Vitroids were used as Gram-negative bacterial strains. The inhibitory effect of our complex polymeric compound on C. albicans WDCM 00054 Vitroids 64 showcased its antifungal activity. These studies were conducted by agar well diffusion 65 (AW) and disc diffusion (DD) studies. The latter consisted of dilution series in comparison to the common antibiotics gentamicin, streptomycin, cefotaxime, chloramphenicol 67 and nystatin. [Cu(12-crown-4)5(H2O)6]I6xI2 exhibits excellent activity against reference 68 strains of microorganisms compared to selected antibiotics” in several different forms is repeated too much in the manuscript. The detailed experimental procedure should be specified in the experimental section and should not be duplicated anywhere else. Thank you very much, agreed. We shortened it in the introduction (lines 58-64) and in Results and discussion: 3.4 Determination of antimicrobial…(first paragraph).. and left this description only in the experimental section.
- The complex consists of sandwiched copper-hexahydrates within polymeric pentaiodide-chains. A recent database check revealed, that the interesting topology of this chain-type polyiodide is new. – A detailed address of the CSD (citation is also absent!) version and search details should be specified so that the readers and reviewers can test what the authors mean by “unique topology”. Thank you, we did some explanations on lines (378-399).
- cupper iodide – incorrect spelling. Yes, changed to copper iodide, in the first line of materials and methods. Thanks.
- X-ray diffraction (XRD) and x-ray crystallography – X-ray with the capital letter and also the fragment may be considered wordy: one or the other part can be omitted. Thanks! We removed the x-ray diffraction.
- Mo-Ka with a wavelength of 0.71073 – the unit of length is absent Sorry, is added now into Table 3.
- Hydrogen atoms were addded – incorrect spelling. True, corrected.
- [Cu(H2O)6(12-crown-4)5]I6 x 2I2 – no upper and lower text indices Yes, thanks, corrected on 2.4
- The 1H- and the 13C- nuclear magnetic resonance (NMR) measurements are in accordance with previous results for similar 12-crown-4-polyiodide complexes with singlets at 70.40 ppm and 3.521 ppm, respectively (Figure 2) [28,30,32]. – seems that the ppm values are given inconsistently: at the beginning of the phrase the sequence is 1H – 13C and the shifts are given so that the 13C values are in the first place. True, yes changed places now.
In my opinion, Fig. 2 should be better replaced to the supplementary section. Agreed and moved.
- Many times in the text there is a spelling "12-Krone-4, [M(12-Krone-4)2]Ix" which in my opinion is incorrect. My bad mistake, sorry. I replaced it with 12-crown-4.
- a peak at 2855 – the unit is absent Unit cm-1 added. Thanks!
- Raman spectrum should be rescaled or a break should be inserted in the medium-wavenumber region, as the main bands discussed by the authors are located below 200 cm-1 and thus interpretation is ambiguous. As the authors kindly provided their raw data, I was able to analyze it carefully. Unfortunately, the laser length used for spectrum excitation induced high luminescence in the most important range. I understand that in many cases the choice of the experimental setup is limited, but still, according to the quality of the spectrum, the authors can make reliably only the assignment of the band of bound iodine. The attribution of the bands at 232 and 351 cm-1 to overtones when we do not see initial bands (which should be intense according to the literature data) maybe too risky – we can also deal with the librations of crown part and anion…cation with respect to each other. Thus, I would recommend replacing Table 1 to the supplementary section and rewriting this part. Dear Reviewer, can we keep Table in the manuscript without moving it to the supplemental material ? Because we show now only the range between 100 and 300 cm-1, the other parts are not visible for the 12-crown-4. Therefore, we think, it may be better to keep the table. Furthermore, we can assign the band at 142 cm-1 to the pentaiodide units in the provided spectrum. Unfortunately, we had no other choice of Raman laser. Thank you for your understanding and support in explaining the situation. We added explanation in lines in lines 267-280 and changed the figure to make it clearer. Thank you so much for your valuable and sincere comments.
- What does the line “12-crown-4 2.891” in table 1 mean? Oh, thanks. It is the Raman shift for 12-crown-4. Mistake was corrected to 2855 cm-1
- What is the meaning of the red color in table 2? Removed, it was needed for the discussion and we forgot to remove the red color. These bands are related to the hydroxyl groups originating from the hydration.
- δ (C–C)s - in general the term symmetric-asymmetric is attributed to stretching vibrations, not deformational ones; True, thanks. Removed from table 2 !
- The crystal structure analysis of [Cu(H2O)6(12-crown-4)5]I6 x 2I2 revealed a novel topology – details of CSD analysis should be added. Please refer also to the added explanations on lines (378-399).
- Figure strukt 1 – unknown notation; Sorry, by mistake. Replaced with Figure 3.
- Cu-O3WÅ: 2.298(3) – the unit is not in its place, Thanks, corrected and moved behind the number.
- Figure 6. [Cu(12-crown-4)5(H2O)6]I6xI2. All shown atoms are drawn with a 50% probabilty of the 374 displacement ellipsids. The branched chain is located parallel to the bc plane running along the c 375 direction – typo in the "ellipsoid" spelling. Corrected, thank you so much. Figure 6 was changed to number 4, because the other reviewers asked us to move some figures into the supplemental material.
- I3- anion – no upper-lower indices. Corrected, thanks
- The presences – typo. Corrected to “presence“
- [49 and references cited there] – unknown citation Thank you, changed to [49].
- The less negatively charged fungal and Gram-positive bacterial species – the phrase is misleading, I guess the authors mean membrane potential. Yes, thank you. The sentence is really too complicated and misleading. Therefore we changed it to : Gram-negative pathogens have higher negatively charged cell surface and are less susceptible towards [Cu(12-crown-4)5(H2O)6]I6xI2.
- membrane fatty acids [] – empty citation, Yes, thanks. It is [53].
Overall, all the comments above do not significantly lower the impression from the manuscript. As a suggestion for the future, it can be very interesting to study the processes of iodine release in details: kinetics, thermal stability, the energy of binding, the stability of powder samples on storage and in solutions, and many other thermodynamic and kinetic parameters that can be interesting from fundamental, physico-chemical and applied bactericidal point of view.
Dear Reviewer, thank you for your sincere, interesting and important comments, which shed for us light into several points. We thank you for your contribution and the efforts to improve this manuscript. The last mentioned points about thermodynamic studies are really interesting and noteworthy. Unfortunately, we are not able to do such studies. Nevertheless, they are relevant to understand the basic parameters, including the mechanisms involved in the interaction between iodine and the microorganisms. I really asked several computational groups to work in a collaborative means, but, it seems, this is not easy to do, given the amount of work and efforts needed to acquire knowledge about polyiodides and microbiology (microorganism morphology) at the same time. I understand that.
One of our reviewers asked us to remove EDS, NMR and FTIR spectra from the manuscript and add them into the supplemental material. We followed that advice.
I really personally enjoyed doing the corrections based on your valuable comments.
Thank you so much.
With our best regards
Zehra
13.3.2021

Reviewer 3 Report
Reviewers' comments:
Manuscript Number: polymers-1146821
Full Title: Antimicrobial hexaaquacopper(II) complexes with novel Polymeric iodide-chains.
Comments:
I recommend accepting it with the following major changes that can complement the text content:
- The introduction section should be improved; more related papers must be discussed and superiority, novelty, critical improvement in this study must be clarified.
- The experimental section should be detailed especially for the 2.3. Characterization of the Compound, 2.3.2. Nuclear Magnetic Resonance (NMR) Spectroscopy, 2.3.3. Characterization by Raman Spectroscopy (Raman), and 2.3.4. Fourier-Transform Infrared (FTIR) Spectroscopy.
- Figure 1. Scanning electron microscopy (SEM) (a) and Energy dispersive spectroscopy (EDS) (b) of 231Cu(12-crown-4)5(H2O)6]I6xI2 - is not clear make clear.
- In part SEM: how the energy of the accelerator beam used?
- Figure 2. Nuclear Magnetic Resonance (NMR) spectroscopic analysis of [Cu(12-crown-4)5(H2O)6]I6xI2. (a) 13C; (b) 1H - is not clear make clear.
- Figure 3. Raman spectroscopic analysis of [Cu(12-crown-4)5(H2O)6]I6xI2 - is not clear make clear.
- The conclusion part should rebuild to let it fluent.
- References: there are recent references in 2019 and 2020 treating the same subject, you can use and make all references in same format for volume number, page numbers and journal name, because it is difficult to searching and reading.
- Language needs substantial improvement. Please consult a native English speaker or a language editing service.
Based on these, I advise the authors to rectify the above-mentioned errors and we hope to re-evaluate the revised manuscript.
Author Response
Dear Reviewer,
thank you for your kind efforts and valuable comments.
Please see the attachment or read below.
Best regards
Zehra
Comments:
I recommend accepting it with the following major changes that can complement the text content:
- The introduction section should be improved; more related papers must be discussed and superiority, novelty, critical improvement in this study must be clarified. Dear Reviewer, thank you for your comment. This type of complex is not common and not well known. Especially, there are only very few publications related to 12-crown-4 or the newly discussed topology in our title compound. We did not enter new references here, but we decided to enrich the discussion, when the structure of the compound is compared to other literature data. By doing this, we followed the comments of the other reviewer (lines 378-399).
- The experimental section should be detailed especially for the 2.3. Characterization of the Compound, 2.3.2. Nuclear Magnetic Resonance (NMR) Spectroscopy, 2.3.3. Characterization by Raman Spectroscopy (Raman), and 2.3.4. Fourier-Transform Infrared (FTIR) Spectroscopy. See comment below**
2.3.3. Characterization by Raman Spectroscopy (Raman)
“The excitation of the solid-state laser beam was focused on the sample through the 50× objective of a confocal microscope with a spot diameter of —2micron). The sample solution was placed into a standard cuvette (1 cm × 1 cm) on the pathway of the laser beam. The CCD-based monochromator collected and analyzed the scattered light.” Was added as information.
- Figure 1. Scanning electron microscopy (SEM) (a) and Energy dispersive spectroscopy (EDS) (b) of 231Cu(12-crown-4)5(H2O)6]I6xI2 - is not clear make clear. See comment below**
- Figure 2. Nuclear Magnetic Resonance (NMR) spectroscopic analysis of [Cu(12-crown-4)5(H2O)6]I6xI2. (a) 13C; (b) 1H - is not clear make clear. See comment below **. Also explanation added: ” Opposite to the previously investigated complexes [M(12-crown-4)2]Ix, the title compound shows a similar 13C –shift like pure 12-crown-4 with 70.40 ppm compared to 70.387 ppm, respectively [28,30,32]. This result verifies, that the 12-crown-4 molecules in of [Cu(H2O)6(12-crown-4)5]I6 x 2I2 are less deformed by the complexation. The 13C-NMR spectra of the sandwiched complexes [M(12-crown-4)2]Ix reveal signals in the range below 70 ppm [28,30,32].” In the lines 253-258.
- Figure 3. Raman spectroscopic analysis of [Cu(12-crown-4)5(H2O)6]I6xI2 - is not clear make clear.
We added explanation in lines in lines 267-280 and changed the figure to make it clearer. Thank you for your valuable comments.
**Dear Reviewer, we outsourced the analysis methods in different private laboratories in India. We contacted them to give us the needed information, but they refused to cooperate, indicating, that it is not necessary to disclose those informations. Our call was rejected and we are unable to contact anyone else regarding this matter. Because they are commercial labs, thay do not consider our interest, once they are paid, the job is done for them at that’s it. We are very sorry for this problem. Actually, it is a very bad experience for us as well and we will surely never repeat again, asking commercial labs to do this by outsourcing. But our own possibilities in our Universities were limited. We needed to find a solution to continue our reseach. At the same time, the labs in India, which did for us our measurements and the university itself is having a partial lockdown. We are not able to do new measurements due to lack of appointments and lack of 12-crown-4. The crown ether delivery did not materialize, because of the same reasons. This is valid for all our analytical methods you mention, which are unclear. Please consider this problem. Since last year, we are fighting the same lockdown problem in India. We moved some of the figures to the supplemental material instead. Thank you for your kind understanding.
- In part SEM: how the energy of the accelerator beam used? We added, that the high-vacuum mode was conducted at 15 kV. Because of the lockdown in India, we could not reach the research assistants in the analytical lab, who did the measurements for us. The same reasons are valid for the Experimental section in the comments you mentioned above. We tried to contact them and could not get any answer within the 10 days given to us to answer the reviewer comments.
- The conclusion part should rebuild to let it fluent. Thanks for the comment, we removed unnecessary details and made it up to the point.
- References: there are recent references in 2019 and 2020 treating the same subject, you can use and make all references in same format for volume number, page numbers and journal name, because it is difficult to searching and reading. We worked on the references section and discovered mistakes, which we all corrected. Thank you for your understanding and your kind support
- Language needs substantial improvement. Please consult a native English speaker or a language editing service. Thank you. We edited the text and tried to remove mistakes.
Based on these, I advise the authors to rectify the above-mentioned errors and we hope to re-evaluate the revised manuscript.
Dear Reviewer, thank you for your kind and valuable comments. We are presenting you the results, which we were able to change. Unfortunately, we received no any support from our outsourced, commercial labs operating from India due to the above explained reasons. Please, kindly understand our problem.
Best regards
Dr. Zehra Edis
13.3.2021
Round 2
Reviewer 1 Report
Accept
Author Response
Dear Reviewer, thank you so much !!!!
Reviewer 2 Report
Dear Authors,
In general, I approve the performed corrections and answers to the questions. The Authors are free to leave Table 1 in the main text if they think that it is essential for understanding the material.
These are some additional remarks:
-In my opinion, the numbers in brackets in table 2 are misleading, as they are very similar to the uncertainty range that is usually presented this way, for example, for bond lengths in X-ray crystallography. Thus, I think a space between the number and opening bracket will help to avoid this unnecessary association. (for example, 3746 (1)).
-The supplementary section now does not include cif file, I think it should be definetely added.
-The authors should more attentively treat the brutto-formula that they attribute to triiodide: in some places it is written as [Cu(H2O)6(12-crown-4)5]I6 x 2I2 while somewhere it is written as [Cu(H2O)6(12-crown-4)5]I6 x I2 (for example, lines 59, 70 and many more)
- In my opinion, conclusion sections may be slightly enriched with the summarization of experimental data, in the present form it is rather brief, but it is a suggestion and is up to the authors.
Author Response
Dear Reviewer,
thank you for your comments.
Best regards
Zehra
The document is attached. Or see also below:
Dear Authors,
In general, I approve the performed corrections and answers to the questions. The Authors are free to leave Table 1 in the main text if they think that it is essential for understanding the material. Thanks a lot!
These are some additional remarks:
-In my opinion, the numbers in brackets in table 2 are misleading, as they are very similar to the uncertainty range that is usually presented this way, for example, for bond lengths in X-ray crystallography. Thus, I think a space between the number and opening bracket will help to avoid this unnecessary association. (for example, 3746 (1)). Dear Reviewer, thank you for this comment. Actually, after removing the figure to the supplementary materials, we also felt, that this is quite misleading and were thinking to remove the numbers entirely. When we tried to add the space between bracket and number, the whole table got disfigured, therefore, we just removed all numbers, even in the text, we removed. I think, now it is much better.
-The supplementary section now does not include cif file, I think it should be definetely added. Yes, true ! I thought I uploaded it, will definitely try to do it. While uploading I had some issues and had to send some figures and documents to the email of the editorial office. I will again ensure, that it is sent to the editorial office. It seems, it doesn’t accept that file type.
-The authors should more attentively treat the brutto-formula that they attribute to triiodide: in some places it is written as [Cu(H2O)6(12-crown-4)5]I6 x 2I2 while somewhere it is written as [Cu(H2O)6(12-crown-4)5]I6 x I2 (for example, lines 59, 70 and many more). Oh, my bad mistake, so sorry for this! I changed everything wrong. Thank you so much !!!!! I changed all to [Cu(12-crown-4)5(H2O)6]I6 x 2I2.
- In my opinion, conclusion sections may be slightly enriched with the summarization of experimental data, in the present form it is rather brief, but it is a suggestion and is up to the authors. Dear Reviewer, actually it was more explnanation, then two reviewers asked us to shorten and we did. Now, the other two reviewers are fine and I do not know, what will happen, if I am going to change again. Sorry for this, I think it is better to keep it as it is.
Dear Reviewer, thank you for your comments, which were again justified. We tried to follow those, which we can change and the tmanuscript is now much better. I will take care of the cif file definitely. I will not upload all in one folder but just try to upload the cif file only. Anyhow, all the other files are uploaded now. We added also the supplementary material information on lines 527-532.
Again, with our compliments and thanks for your efforts in improving our work.
Best regards
Zehra
15.3.2021

Reviewer 3 Report
Reviewers' comments:
Manuscript ID: polymers-1146821
Full Title: Antimicrobial hexaaquacopper(II) complexes with novel Polymeric iodide-chains.
The authors revised the manuscript according to the reviewers' comments.
So that I recommended this manuscript accept for publication in Polymers.
Author Response
Dear Reviewer, thank you so much for your kindness !!!